# Lifestyle and Health-Related Quality of Life Relationships Concerning Metabolic Disease Phenotypes on the Nutrimdea Online Cohort

**DOI:** 10.3390/ijerph20010767

**Published:** 2022-12-31

**Authors:** Andrea Higuera-Gómez, Rosa Ribot-Rodríguez, Victor Micó, Amanda Cuevas-Sierra, Rodrigo San Cristóbal, Jose Alfredo Martínez

**Affiliations:** 1Precision Nutrition and Cardiometabolic Health, IMDEA-Food Institute (Madrid Institute for Advanced Studies), Campus of International Excellence (CEI) UAM+CSIC, 28049 Madrid, Spain; 2Centre Nutrition, Santé et Société (NUTRISS), Institut sur la Nutrition et les Aliments Fonctionnels de L’Université Laval (INAF), Université Laval, Quebec, QC G1V 0A6, Canada; 3School of Nutrition, Université Laval, Quebec, QC G1V 0A6, Canada; 4CIBERobn Physiopathology of Obesity and Nutrition, Institute of Health Carlos III (ISCIII), 28029 Madrid, Spain

**Keywords:** metabolic health and disease, Mediterranean diet, physical activity, health-related quality of life, online data collection

## Abstract

Obesity, diabetes and cardiovascular events are non-communicable diseases (NCDs) directly related to lifestyle and life quality. Rises on NCDs rates are leading to increases in early deaths concerning metabolic morbidities. Health-related quality of life (HRQoL) has been described as a subjective perception about the influence of health and personal features on human well-being. This study aimed to characterize phenotypic and lifestyle roles on the occurrence of metabolic diseases and determine the potential mutual interactions and with HRQoL. Data from an online adult population (NUTRiMDEA study, *n* = 17,332) were used to estimate an adapted Obesogenic Score (ObS), while logistic regression analyses were fitted in order to examine relevant factors related to the prevalence of different metabolic diseases including HRQoL. Sex and age showed significant differences depending on lifestyle and metabolic health (*p* < 0.05). Adherence to the Mediterranean diet and physical activity showed a mutual interaction concerning ObS (*p* < 0.001), as well with metabolic health (*p* = 0.044). Furthermore, metabolic diseases showed own features related to sociodemographic and lifestyle characteristics in this population. Metabolic syndrome components may be differently influenced by diverse lifestyle or socioeconomic factors which in turn affect the perceived HRQoL. These outcomes should be taken into account individually for a precision medicine and public health purposes.

## 1. Introduction

An increase in premature deaths and years of life lost are associated to the growing rates of non-communicable diseases (NCDs) such as obesity, diabetes, or cardiovascular diseases [1,2,3]. Body mass index (BMI) is a commonly used quantitative measure of adiposity. Obesity refers to a BMI exceeding the 95th percentile and is associated with type 2 diabetes mellitus, hypertension and hyperlipidemia [4]. The main risk factors of NCDs can be classified into genetic, medical conditions, environmental (diet, physical activity), and sociodemographic features [5]. Furthermore, sex and age are important determinants that impact cardiometabolic features [6]. Worldwide NCDs kill 41 million people each year, equivalent to 74% of all death globally [7]. Specifically, in Spain, NCDs are the main cause of death, with 388,617 corresponding to 92.8% of all deaths [8]. Not only genetic and age is related with disease, unhealthy dietary habits, physical inactivity, sleep deprivation and smoking are modifiable lifestyle habits and behaviors that have been recognized as contributors to the onset of NCDs [9,10]. However, it is also important to consider the interactions that can arise between them [11]. Metabolic syndrome (MetS) refers to a conglomerate of clinical manifestations [12] that include obesity or high waist circumference, hyperglycemia, high blood pressure (HBP), and dyslipidemia as elevated triglyceride levels and/or low high-density lipoprotein cholesterol levels [13,14,15], which are associated with a 4–5-fold increase in type 2 diabetes and a 2–3-fold increase in cardiovascular disease (CVD) [14,16].

The impact of lifestyle on people’s health has been described [17,18]. Thus, dietary patterns have been related to health [19,20,21,22,23]. Thus, the Mediterranean diet has shown to promote the prevention of some NCDs [21,24]. The beneficial effects of this dietary pattern have been mainly attributed to the anti-inflammatory and antioxidant properties provided by the foods consumed [10,25,26]. Physical activity is another factor promoting health and preventing the onset of NCDs [27,28], as well as a therapeutically agent [29,30,31].

As stated by World Health Organization (WHO) “Health is a state of complete physical, mental and social wellbeing and not merely the absence of disease or infirmity” [32]. The concept of ‘well-being’ recognizes three broad spheres: subjective well-being, psychological well-being and social well-being [33], which is associated to health-related quality of life (HRQoL) and has been described as a subjective perception about the influence of health and personal features on human life [34]. There are different validated tools for measuring HRQoL, such as European Quality of Life-5 Dimensions (EQ-5D), WHO Quality-of-Life Scale (WHOQoL) [35], Nottingham questionnaire [36], Short-Form 36 (SF-36), or Short-Form 12 (SF-12) which is a validated subset of SF-36 that measures the same eight domains and yields a Physical Component (PCS12) and a Mental Component (MCS12) [37,38]. A categorization of nutritional wellbeing and health status into qualitative (proto)nutritypes led to develop a pilot quantitative nutrimeter to objectively integrate nutritional information, which evidenced the association with quality of life [39].

In order to devise valid tools more focused on both prevention and treatment of the disease, it is necessary to categorize people depending on accompanying comorbidities [40]. Online data collection and the use of digital technologies for the implementation of novel health strategies and interventions have gained popularity due to it may offer additional advantages over conventional approaches [41,42]. For this purpose, different indices have been developed, such as the Charlson index with a wide clinical evidence-based utility [43]. The Obesogenic Score (ObS) is an index to determine the role of lifestyle and health factors [44,45].

This study aimed to determine not only which sociodemographic features, phenotypic traits and lifestyle habits are important in health and metabolic disease, but specifically the potential interactions between them involving health/disease outcomes and the relationships with HRQoL in an online recruited adult population.

## 2. Materials and Methods

### 2.1. Study Design

The NUTRiMDEA survey included a total of 17,332 online participants who were screened online from May 2020 to November 2020 (Figure 1). The baseline questionnaire collects information on socio-demographic data, metabolic history, anthropometric variables and lifestyle aspects. In order to determine the nutritional status or nutritypes (qualitative nutritional categorization) that can metabolically classify healthy and sick people and the role of different environmental factors related to nutritional well-being and subsequently issuing specific recommendations for each of them. The survey was based on previously validated questionnaires including phenotypic, lifestyle, HRQoL and well-being features [38,39]. The questionnaire was administered in two different settings. The first scenario, an open survey (OS) through a web-based platform tailored for online surveys (QuestionPro), was freely accessible online at https://nutrimdea2020.questionpro.com/ (accessed on 1 May 2020). The second scenario involved that, audiences were purchased from different online survey tools (Survey Planet, QuestionPro panel, Zoho Survey, Netquest). In this case, respondents are rewarded by the chosen platforms (RS). The variables collected in the study were age, gender, ethnicity, education, home situation, work status, weight, height, pant and shirt size, metabolic disease diagnosis (obesity, diabetes, HBP, dyslipidemia), familial metabolic disease diagnosis (obesity, diabetes, HBP, dyslipidemia), sad or depress feeling, sleep routine, smoke habit, number of meals per day, snacking habit, water consumption, added salt in meals, SF-12 survey and finally diet and physical activity questionnaires are described below. Further characteristics of the sample were described elsewhere [46].

### 2.2. Physical Activity

Physical activity and sitting time information were collected by administrating the International Physical Activity Questionnaire (IPAQ) validated for Spanish population [47,48], which allowed to estimate light, moderate and intense activity minutes per week. Total METs per week (METs-min/week) were also calculated with such a questionnaire.

### 2.3. Mediterranean Diet

Another section of the survey encompassed the 14-item Mediterranean Diet Adherence Screener (MEDAS-14) from the project Prevention with Mediterranean Diet (PREDIMED), which collects the consumption of olive oil, vegetables, fruits, legumes, poultry, red and processed meats, butter, beverages, red wine, fish, pastries and nuts. The final score ranges from 0 to 14 points [49,50].

### 2.4. Obesogenic Score (ObS)

The ObS was obtained by a modified version of the obesogenic behavior score [44]. This tool gives a cumulative score for different behaviors: +1 point for low physical activity (<1 h/day), +1 point for being ≥2/day in front of screens (sedentary), +1 point for skipping breakfast and +1 point if <3 meals/day. The final score ranges from 0 to 4 points, with higher scores indicating an accumulation of unhealthy obesogenic behaviors [44]. The modified version score for this study was calculated by giving +1 point for low physical activity (≤150 min/week) according to WHO recommendations for adults [51]; +1 point for sedentary lifestyle (≥8 h/day) which has been shown to increase the risk of mortality [27,52]; +1 point for skipping meals for snacks; +1 point for ≤3 meals/day. Participants scoring 3 and 4 in the ObS were combined as a single group (≥3 points) due to underpopulation in these 2 groups separately (*n* = 2070 and *n* = 362, respectively).

### 2.5. Statistical Analyses

Conventional descriptive statistics were performed for comparisons of sex (men and women) and age (<40 years and >40 years), in relation to sociodemographic, health or disease parameters and lifestyle criteria (physical activity, sedentary lifestyle, sleep and eating habits, including MEDAS-14 score) as relevant variables in this study. Descriptive statistics were also performed to analyze the differences in the aforementioned variables by type of survey (open and rewarded), physical activity and Mediterranean diet, the latter two stratified dichotomously using the median of each variable as cut-off value. The characteristics of the sample relative to the distribution by sex and age were tabulated and statistically analyzed with T-student test to compare parametric continuous variables and chi-square (χ^2^) test for categorical variables described as proportions, to examine the possible variables required for the subsequent regression analyses. Median and interquartile intervals were analyzed by Kruskal-Wallis test for non-parametric variables light, moderate, intense and total PA.

Differences between dichotomous groups and the presence of interactions with ObS were analyzed by analysis of variance (ANOVA) and the Tukey HSD test for posthoc analysis. It was adjusted for confounding factors such as age, sex, type of survey, HRQoL (PCS12 and MCS12) and education. In addition, interaction between healthy status or presence of metabolic diseases and adherence to Mediterranean diet with physical activity was analyzed by analysis of variance (ANOVA) and the Tukey HSD test for posthoc analysis. It was adjusted for confounding factors such as age, sex, type of survey, HRQoL (PCS12 and MCS12) family disease, sedentarism and mean meals per day. As well as the interactions between the type of survey and physical activity, and between the type of survey and the Mediterranean diet in the NUTRiMDEA population. Differences were studied with two-way ANOVA and the Sidak test for post hoc analysis. It was also adjusted for confounding factors such as age, sex, type of survey and HRQoL (PCS12 and MCS12).

Logistic regression analysis was performed to investigate the association between health and disease (obesity, diabetes, HBP, dyslipidemia, separately) and dietary, sociodemographic, physical activity, and HRQoL.

All the performed statistical tests were two-tailed, and *p*-values < 0.05 were considered statistically significant. Statistical Analyses were performed using RStudio version 4.0.1.

## 3. Results

Significant differences were found according to sex (female/male) comparison for most of the variables, except for work status, family history of obesity, PCS12, water consumption habits and light physical activity. More specifically, women exhibited to have a lower MCS12 score, declared less practice of moderate, intense and total physical activity (METs-min/week) and were more sedentary than men. Nevertheless, women presented a higher Mediterranean diet score (MDS14) (*p* < 0.001) and a lower BMI (*p* < 0.001). Women declared a lower prevalence of metabolic diseases (diabetes, obesity, HBP and dyslipidemia) than men, but women reported a greater number of family metabolic diseases (Table 1 and Table 2).

Further statistical differences were found when the sample was categorized by age: <40 years old vs. >40 years old. As expected, older individuals had a greater prevalence of metabolic diseases and a higher BMI (Kg/m^2^) (Table 1). Meanwhile, a higher overt low mood or overt depression (*p* < 0.001) was observed in the population aged <40 years, which coincides with a lower score in MCS12 (*p* < 0.001). However, this population presented a higher score in PCS12 compared to the >40 years group (*p* < 0.001) (Table 1). Interestingly, younger individuals showed a lower MDS14 (*p* < 0.001), decreased levels of light physical activity, but more moderate, intense and total physical activity (*p* < 0.001 for low, moderate and high and *p* = 0.037 for total physical activity), as well as more sedentary time during the week (*p* < 0.001) (Table 2).

Regarding the analysis of interactions between diet and physical activity, significant differences were seen for the MDS14 and ObS (Table 3). Higher ObS was found in those individuals with the lower MDS14 and physical activity (dichotomous) and inverse trends when both MDS14 and physical activity were high (Appendix A).

Greater differences were observed for MDS14 (between dichotomous groups) in the low physical activity group compared to the high physical activity group (Table 3). A higher gap in the MDS14 was also found in the low MDS14 group compared to the high MDS14 group in relation to high or low physical activity (Table 3). The low MDS14 group, which had a higher physical activity, was related to a greater increase in MDS14, compared to the high MDS14 group and a high physical activity. Interaction between both variables was noted (Appendix A).

Differences in family history, HRQoL (PCS12 and MCS12), BMI, ObS and lifestyle of the participants in the presence or absence of metabolic diseases were studied (Table 4). Noteworthy, significant differences were not observed between the participants reporting obesity for sedentarism (*p* = 0.976), with diabetes prevalence for physical activity (*p* = 0.047) and ObS (*p* = 0.716), regarding the presence of HBP for MDS14 (*p* = 0.932) and ObS (*p* = 0.609), and with respect to dyslipidemia for MCS12 (*p* = 0.053), sedentarism (*p* = 0.119) and ObS (*p* = 0.977) (Table 4).

On the other hand, significant differences were found in the proportion of family disease history for each disease, being higher in HBP and lower in obesity (Figure 2). Regarding BMI (Kg/m^2^), the group with obesity showed the highest level and the group with dyslipidemia the lowest level (*p* < 0.001). There were significant differences in the physical component of HRQoL (PCS12) between participants with dyslipidemia and the rest of the diseases, who obtained a lower score (*p* < 0.001). Regarding the mental component of HRQoL (MCS12), significant differences were found between participants, with the lowest score in patients with obesity and the highest score for patients with HBP (*p* < 0.001). The participants who declared obesity or diabetes obtained a lower score in the MDS14 compared to the participants who declared HBP or dyslipidemia (*p* < 0.001). Regarding sedentary time, the participants who declared obesity or dyslipidemia displayed the highest levels (*p* = 0.003). No significant differences in physical activity were found between the metabolic disease groups (*p* = 0.079). The group of individuals presenting obesity obtained the highest score in ObS, followed by the diabetes group (*p* < 0.001).

The logistic regression analyses for presence of metabolic diseases showed that there was a significant association between family disease and age (Table 5). Sex also showed significant outcomes in diabetes, hypertension and dyslipidemia, although not in obesity. The type of survey also showed significant differences for obesity, diabetes, and HBP, but not for dyslipidemia. The HRQoL, in terms of the physical component (PCS12), showed a relationship with the four metabolic diseases. Regarding the mental component (MCS12), it showed a relationship with obesity and dyslipidemia, but not with diabetes or HBP. The Mediterranean diet showed a relationship only with obesity, not for the rest of metabolic diseases. Sedentary time seemed to influence only HBP. Contrary to expectations, physical activity (METs-min/week) did not show significant differences with any metabolic disease. Smoking habit showed significant differences only with those reporting dyslipidemia (Table 5).

Finally, when diet and physical activity were compared in the presence or absence of diseases (Figure 3), an interaction was found between the Mediterranean diet and the absence or presence of diseases associated with the level of physical activity practiced. Physical activity was higher when there was a high Mediterranean diet and health followed by 1 disease and ≥2 diseases. In addition, less physical activity is performed when the Mediterranean diet is low, regardless of the absence or presence of diseases.

Regarding the interaction between the type of survey and physical activity (dichotomous), significant differences were found in light, moderate and total physical activity, as well as in MCS12 and ObS (Appendix A). Furthermore, an interaction between the type of survey and the Mediterranean diet (dichotomous) was found, where there were significant differences in MDS14 and PCS12 (Appendix A).

## 4. Discussion

This online cross-sectional study showed that metabolic syndrome components may be differently influenced by diverse dietary, lifestyle or socioeconomic factors which in turn are related with the perceived HRQoL, as previously described in other studies [6,53,54,55,56]. Interestingly, diet and physical activity showed an interaction between them concerning ObS, which is a major finding of this investigation. This study contributes to characterize phenotypic and lifestyle roles on the occurrence of metabolic diseases and HRQoL as well as demonstrate potential interactions between some of them in order to contribute for a precision nutrition and medicine and public health purposes.

In a previous subanalysis the data of the NUTRiMDEA population were compared with the available data at that time of the Spanish National Health Survey (2017), finding as compared with the Spanish general population, that the NUTRiMDEA population had a better cardiometabolic health with a lower prevalence of obesity, HBP, diabetes and dyslipidemia, as well as fewer smokers, better self-perception of their health, lower BMI, greater consumption of vegetables per day and no significant differences in light physical activity [57]. The prevalence of metabolic diseases of this study compared with the European Health Survey in Spain (EESE) of 2020 was 5.1% vs. 16% of obesity, 3.1% vs. 7.5% diabetes, 8.6% vs. 19% of HBP and 15.1% vs. 15.3% of dyslipidemia, respectively [58]. The prevalence of MetS is around 30% of the adult population in high-income countries [59], and its heritability has been estimated to range from 20–60% [14]. MetS traits manifest a significant increase with age and, on average, men have a worse metabolic profile compared with women [60] which is in accordance with our data. The importance of detecting MetS is undoubtedly to identify individuals at high risk for CVD as coronary artery disease, as well as associated determinants [15].

The lower prevalence of metabolic diseases, despite the higher family history reported by women is consistent with previous studies that observed that men were less likely to declare a family history of disease compared to women [60,61,62,63]. In this population, a higher prevalence of obesity was found in men than in women, which coincides with the data of the Spanish National Health Survey 2020–2021 [58,64]. Nevertheless, no significant differences were found for the ObS regarding sex in this population, where sex showed low relevance in the regression model for obesity prevalence.

Regarding age, a higher prevalence of obesity was found in the population >40 years of age. However, the ObS was higher in the population <40 years of age and the logistic regression model showed that age was a discriminant factor regarding the prevalence of obesity as described previously [56]. These results might suggest that an early age could counteract bad habits in the development of obesity. In fact, the European prevalence of obesity increases with age to peak at about 60 years, especially among women aged ≥50 years in Spain [56,65].

Data of this study suggested that active people were more conscious of health maintenance. It was found a higher physical activity in the absence of each metabolic disease compared with the presence of each one, with significant differences for all metabolic disease except for diabetes. People more active smoked less, fewer years and less cigarettes, had a higher adherence to Mediterranean diet, ate 3 or more meals per day, had less snacking habit, drank more water and had better sleep routines. Similarly, people with higher adherence to Mediterranean diet also presented a higher physical activity practice. People younger, more students, who lived alone, had a less sedentary lifestyle, with more prevalence of diabetes were more physically active but had a lower adherence to Mediterranean diet. Secondly, older participants, mainly women, more workers, who lived in couple or with children, had a more sedentary lifestyle, with more prevalence of dyslipidemia had a higher adherence to Mediterranean diet but were less physically active.

The categorization of our population based on high or low adherence to Mediterranean diet and physical activity practice allowed us to discern the presence of a group of people who seemed not to pay attention to their habits. The adherence to Mediterranean diet seemed to influence the ObS less when the level of physical activity was high. It could also be interpreted that the Mediterranean diet could influence more when the level of physical activity was low. The same occurred to physical activity, as it seemed to have less influence on the ObS when the Mediterranean diet was high and to have more influence when the Mediterranean diet was low. In other words, the most active people are also more concerned about their diet habits. According to these findings, observational studies report that physically fit individuals are more likely to follow recommended dietary guidelines [66,67]. However, changes in both diet and exercise habits seem to trigger more benefits together than those reached by either alone, but it is also important to take into account the individual genetic makeup to achieve more efficient clinical interventions [14]. Evidence supports the interactions between diet and physical activity as an effective approach to mitigate age-associated cognitive decline [66]. On the other hand, the interaction between diet, physical exercise and psychosocial well-being in women with gestational diabetes mellitus has been reported, indicating that the interaction between these three lifestyle domains produced desirable outcomes [68]. The interaction between diet and exercise with gut microbiota has also been studied [69]. In addition, physical activity gained special relevance due, among other factors, to the potential anti-inflammatory effect [70].

We found that people with chronic diseases were more likely to practice less physical activity (except for diabetes). According with these results, the SCAPIS cohort study with more than 27,000 participants established that the presence of one or more chronic diseases had strong associations with higher sedentary time and less time in all physical activity intensities [71]. Nevertheless, moderate exercise has been reported to be associated with a longer life expectancy, also in individuals with 2 or more chronic conditions [72]. Despite the presence of multimorbidity, adopting a healthier lifestyle was associated with up to 6.3 years longer life for men and 7.6 years for women. In fact, a combination of various healthy lifestyle behaviors, rather than a strong emphasis on a particular one seems to be an optimal strategy to improve cardiovascular health [73]. Not all risk factors related to lifestyle were equally correlated with life expectancy, with smoking being significantly worse than the rest [74]. In our population, smoking was related only to dyslipidemia.

Variables included in the statistical regressions represented different aspects of the characteristics or lifestyles of the participants. Physical activity was not clearly associated with the prevalence of metabolic diseases in this research, contrary with other reports, where there is an inverse association between physical activity and different metabolic conditions [75,76]. In this sense, it could be discussed if the relationships between physical activity and each metabolic disease are due to cause or consequence, which may explain and contribute to interpret this apparent controversy.

Our results might suggest the presence of certain relationships between physical activity and dyslipidemia. These data may be consistent with a previous meta-analysis that compared the effectiveness of pharmacological treatment and physical exercise strategies to improve blood lipid profile in high-risk cardiovascular disease patients [77]. Both exercise and ‘polypill’ strategies were effective in reducing low-density lipoprotein cholesterol (LDL-c) and total cholesterol but only exercise improved high-density lipoprotein cholesterol (HDL-c) and reduced triglyceride levels and provides additional health benefits (e.g., increases in physical fitness and decreases in adiposity). It seemed reasonable to recommend exercise as the first treatment option in dyslipidemia when the patient’s general condition and symptoms allow [77].

Another trial observed that subjects with cardiometabolic disorders presented a lower adherence to the Mediterranean diet [57], while in this study, it has been noticed that MDS14 was not always lower in the presence of metabolic disease (it was even higher in the presence of dyslipidemia). Again, it could be discussed if it is a cause or consequence. Contrary to other studies [56], a tendency to a sedentary lifestyle was not observed in the obese population compared to the non-obese population. Similarly, a sedentary lifestyle (>8 h/day) was more prevalence in the groups of absence of disease. It was also a factor that showed to be related to HBP presence in the logistic regression analysis, it seemed to reduce the risk of HBP. This outcome could be explained as a consequence and not as a cause, that is, people diagnosed with HBP could be less sedentary because they have been prescribed physical activity, or even, the fact might be that a subject who is concerned about suffering from a disease is more likely to become active in counteracting its consequences. Therefore, it may appear a stronger motivation to maintain a healthier lifestyle [73,78]. However, it has also been shown that patients with multimorbidity (coexisting type 2 diabetes and hypertension) generally exhibit poor self-care adherence (diet, physical exercise, use of tobacco or alcohol), which negatively affects their disease control [79]. It was also studies that patients with longer disease duration were found to be more likely to adhere to self-monitoring/self-care [79].

Better perception of HRQoL has been related in previous studies with incremented level of physical activity [57]. These results agree with the findings of our study, which suggest that PCS12 is higher in the healthy than the diseased population. However, the MCS12 did not show the same trends. In this context, another study evidenced that the female sex, older age, higher BMI, less education and the presence of chronic diseases were associated with lower HRQoL scores [53]. Other authors explored the self-perceived health status, finding a progressively deterioration associated with the increase of the BMI and the age [56,80]. Likewise, we found significant differences in HRQoL when population was categorized by age. Similarly, the estimation of PCS12 appeared to be lower in subjects > 40 years, but contrary to expectations, MCS12 is higher in this population. In this regard, it should be noted that the questionnaires were carried out during the COVID-19 pandemic, which may have affected some outcomes [81,82,83,84,85]. Regarding sex, no significant differences were found in the PCS12, but in the MC12, with the female sex being the one that obtained a lower score. Lower PCS12 scores were found in the presence of a metabolic disease (obesity, diabetes, HBP or dyslipidemia). However, contrary to expectations, a higher MCS12 score was observed in people with diabetes compared to people without diabetes and also in people with HBP compared to the population without HBP, and there were no significant differences in the MCS12 score in participants with dyslipidemia vs. non-dyslipidemia. These results suggest that the presence of any metabolic disease may be a limiting factor in physical HRQoL (PCS12) but not in mental HRQoL (MCS12).

High adherence to the Mediterranean diet and an increased physical activity were related to better health perception in this research. These results agree with previous studies [19,53]. Thus, a randomized controlled trial of a 3-months in moderate-to-severe depression, demonstrated a significantly greater improvement in the dietary intervention group and remission achieved in 32% of this group, indicating that dietary improvement may provide an effective and accessible treatment strategy for the management of this mental disorder [9]. Furthermore, increased adherence to the Mediterranean diet is associated with better cognition in adults [86]. Physical activity and cognition have been extensively studied, where observational studies show a lower incidence of cognitive impairment in people who maintain regular physical activity [66]. Based on these evidences, some researchers argue for the inclusion of depression and anxiety as high prevalence non-communicable diseases influenced by lifestyle [87]. Therefore, it is feasible and timely to begin to develop effective, sustainable, prevention initiatives related with lifestyle adaptations. Thus, in a 27-year longitudinal study, obesity in mid-life was shown to double the risk of developing dementia at a later age [88]. It is therefore important to highlight the beneficial effects of the Mediterranean diet and physical activity on metabolic, and mental health [10].

The participants of the two types of surveys (open and rewarded) in this research showed different characteristics, suggesting the survey may have a role in the interaction with diet and physical activity. It can be hypothesized the method of online data collection may influence the characteristics of the participants [46]. Depending on the objective of the study, it is necessary to take into account the methodology in data collection, being relevant the choice of a population sample which is truly representative. In any case, statistical corrections were implemented but the tendencies remained.

Given the background that participants in a prospective observational study indicated a need for personalized advice to facilitate their physical activity practice [89], and dietetic individual advice appears to be effective in improving diet quality and cardiovascular health [90,91], it seems important to tailor diet and physical activity advice to each individual considering their mutual interactions, towards precision medicine, which is a major finding of this research, that demonstrated that both features should be considered altogether to synergize benefits.

The choice of a healthy diet, associated with regular physical activity and avoidance of smoking, seems promising to reduce the inflammation associated with metabolic syndrome and is essential to combating chronic diseases [21]. The trilemma of diet-environment-health is a global challenge, and opportunity, of great environmental and public health importance [23].

## 5. Limitations and Strengths

A limitation of this study is that all data collected were self-reported, therefore, there is a possibility that not all responses are completely accurate [92] and thus, results should be interpreted with caution. Although there is already available literature that validates this type of online methods [93], it has been described both in this research and in other investigations that a proportion of people tend to slightly underestimate their own weight and to increase their height [56] but the validity of the outcomes has been considered acceptable.

The use of an online method for data collection can be criticized for not being representative of the whole population, because it excludes computer illiterate and those without access to the internet [94]. Nevertheless, online collection has been validated in other surveys such as the SUN cohort [95], Food4Me [93], Nurses’ Health study [96] and Health Professionals follow-up study [97,98] or PROM study [42] supporting the validity of findings involving an online recruited population. Furthermore, rewarded platforms have been shown to be more diverse and thus, recruit a wider range of the general population than other web-based and in-person recruitment methods [99]. The influence of the COVID-19 pandemic coincident with data collection cannot be ruled out, so there may biases some data.

Another disadvantage of this cross-sectional study is that it could not study the cause-and-effect relationship between variables. Although it has been possible to analyze the relationships between some variables, their interaction and the degree to which certain variables influence others and could give rise to the hypothesis that could be assessed in future prospective trials, focusing on analyses where inputs and outputs are well controlled.

In terms of strengths, the sample size in this study is considerable. In addition, it includes different data collection methods (open and rewarded survey), which increases the heterogeneity of the sample, which can represent the population to a greater extent. Furthermore, the questionnaire collects a wide range of information with validated tools. It collected anthropometric, sociodemographic, health and metabolic disease, lifestyle, diet, physical activity and HRQoL data which may offer a broader picture of the population facing implement precision nutrition and medicine.

## 6. Conclusions

In conclusion, differences in behavioral and lifestyle habits, sociodemographic and phenotypic traits, and HRQoL were found in the presence of metabolic diseases and health conditions. Diet and physical activity showed interaction concerning ObS, which are complex multidimensional habits that may reinforce each other and influence health and disease. Both determinants should be considered together when implementing health actions.

It is noteworthy that disorders of the metabolic syndrome have distinctive features. This study has been able to discriminate different phenotypes, lifestyle features and HRQoL in metabolic diseases. Given the increasing incidence and prevalence of chronic NCDs, a categorization based on metabolic health and disease traits is essential for personalized nutrition and precision medicine, which must be considered a public issue for prevention, taking into account interactions of diet and physical activity and the connections with HRQoL.

## Figures and Tables

**Figure 1 ijerph-20-00767-f001:**
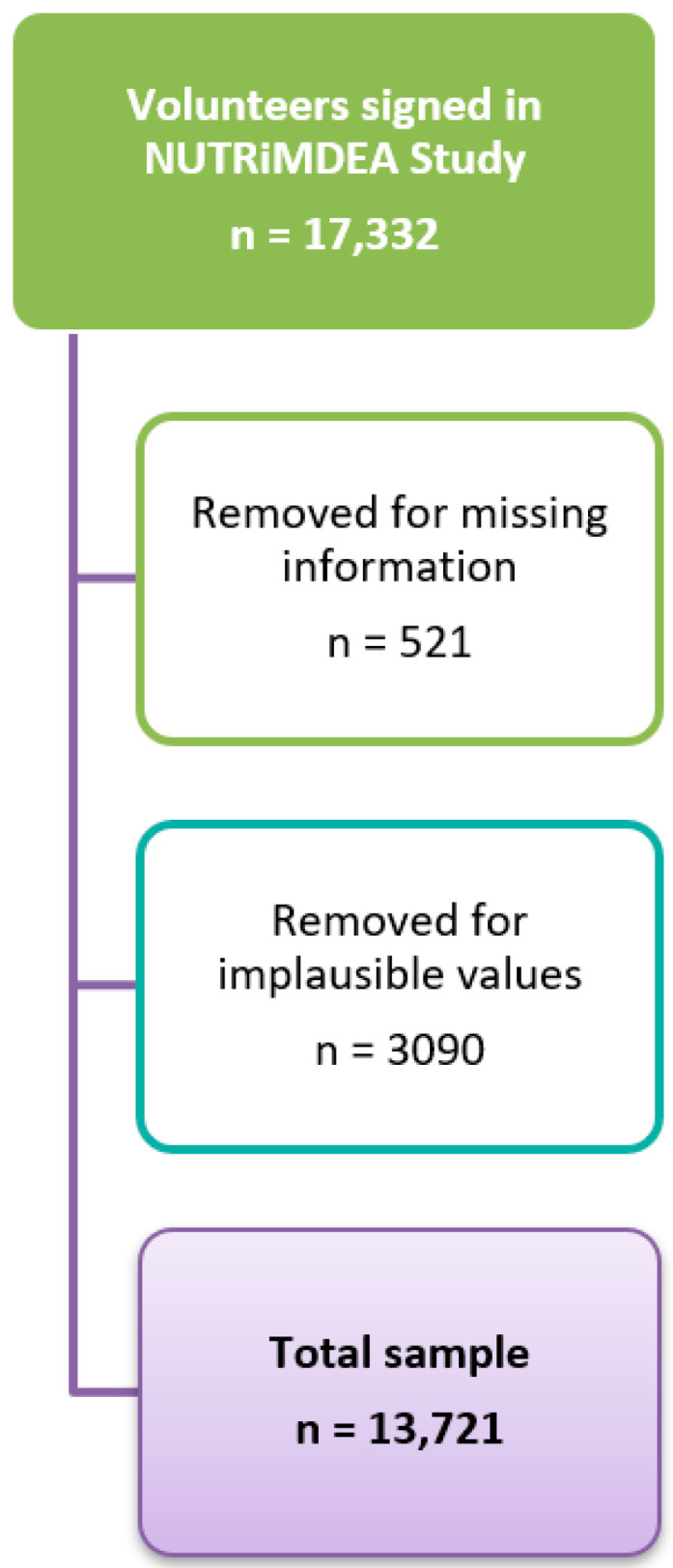
Flow diagram of the NUTRiMDEA Study. The implausible values were outliers which were calculated using boxplots. For weight, the lower limit was set at 19 Kg and the upper limit at 110 Kg. For height, the lower limit was 142 cm and the upper limit 193 cm. For pant size, the lower limit was 28 and the upper limit was 54. For BMI, the lower limit was 13.22 Kg/m^2^ and the upper limit was 34.67 Kg/m^2^. For light PA, the upper limit was from 900 min/week. The upper limit of moderate PA was set at 550 min/week and 600 min/week for PA intense. For total METs, the upper limit was 7194 METs-min/week.

**Figure 2 ijerph-20-00767-f002:**
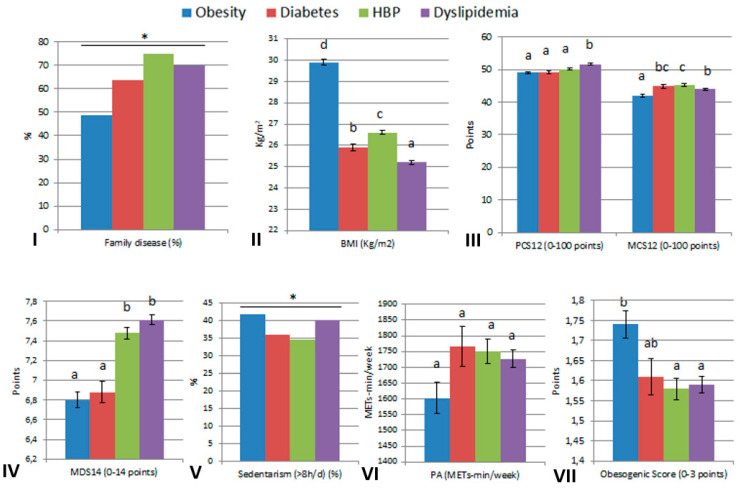
Characteristics of different diseases (obesity, diabetes, HBP and dyslipidemia). (**I**) Prevalence of family disease (%). (**II**) BMI (Kg/m^2^). (**III**) HRQoL score PCS12 and MCS12 (0–100 points). (**IV**) MDS14 (0–14 points). (**V**) Prevalence of sedentarism (%). (**VI**) Physical Activity (METs-min/week). (**VII**) Obesogenic Score (0–3 points). Family disease (%) refers to the same disease in each column. The statistics used to compare mean differences in Family disease (%) and sedentarism (%) were the chi-square test while one way analysis of variance (ANOVA) with sidak post-hoc test was performed for the rest (PCS12, MCS12, Obesogenic Score, MDS14, BMI and PA). Threshold significance was set at *p* < 0.05. * *p* < 0.05. ^abcd^ means with different superscripts are statistically different.

**Figure 3 ijerph-20-00767-f003:**
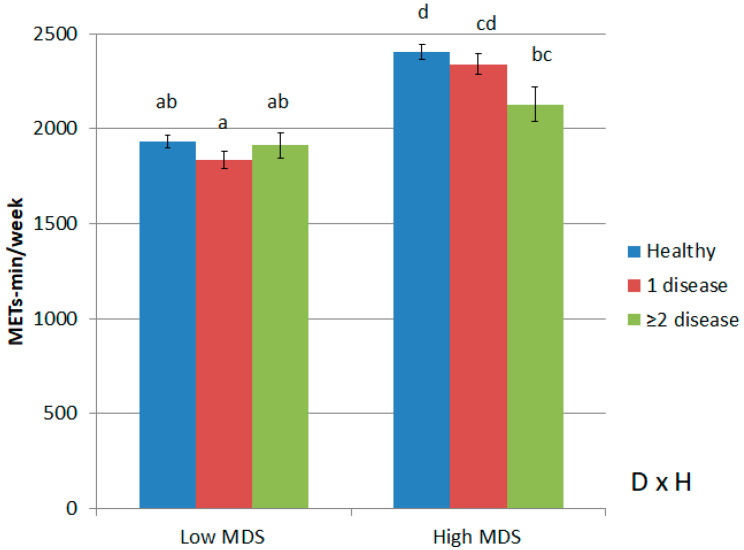
The interaction of healthy status or presence of metabolic diseases and dichotomized Mediterranean diet Score (MDS14) into high and low, in relation to total physical activity (METs-min/week) (mean ± SE). D × H means interaction between Mediterranean diet Score (MDS14) and metabolic Health/disease. Threshold significance was set at *p* < 0.05. *p*-values for two-way analysis of variance (ANOVA) with sidak post-hoc test between groups. Adjusted by type of survey, HRQoL, age, sex, family disease, sedentarism and mean meals per day. Different letters ‘abcd’ indicate statistically differences. Thus, significant differences were observed within the high Mediterranean diet group between health and ≥2 metabolic diseases

**Table 1 ijerph-20-00767-t001:** The characteristics of the participants in the NUTRiMDEA online study according to clinical and sociodemographic variables categorized by sex and age.

		Sex ^a^		Age	
	Overall	Female	Male	<40 Years	>40 Years
*n*	13,721	9031	4690	5335	8386
Age, *n* (%) >40 years	8386 (61.1)	5361 (59.4)	3025 (64.5) *	0 (0.0)	8386 (100.0) *
Sex, *n* (%) of female	9031 (65.8)	9031 (100.0)	0 (0.0) *	3670 (68.8)	5361 (63.9) *
BMI ^b^, mean (SD)	24.07 (3.56)	23.48 (3.62)	25.22 (3.15) *	23.18 (3.45)	24.64 (3.52) *
Trousers size, mean (SD)	40.47 (3.98)	39.75 (3.43)	41.86 (4.56) *	39.14 (3.75)	41.31 (3.89) *
Ethnicity, *n* (%) *					
Caucasian/European	9063 (66.1)	6204 (68.7)	2859 (61.0)	3457 (64.8)	5606 (66.8)
Hispanic/Latino/a	4145 (30.2)	2483 (27.5)	1662 (35.4)	1579 (29.6)	2566 (30.6)
Other ^c^	513 (3.7)	344 (3.8)	169 (3.6)	299 (5.6)	214 (2.6)
Education, *n* (%) *					
Compulsory/Professional Education	3330 (24.3)	1874 (20.8)	1456 (31.0)	1383 (25.9)	1947 (23.2)
Higher education	10,216 (74.5)	7035 (77.9)	3181 (67.8)	3874 (72.6)	6342 (75.6)
Other	175 (1.3)	122 (1.4)	53 (1.1)	78 (1.5)	97 (1.2)
Work, *n* (%) *					
Employed	10,202 (74.4)	6744 (74.7)	3458 (73.7)	3676 (68.9)	6526 (77.8)
Unemployed	2565 (18.7)	1673 (18.5)	892 (19.0)	848 (15.9)	1717 (20.5)
Student	954 (7.0)	614 (6.8)	340 (7.2)	811 (15.2)	143 (1.7)
Obesity, *n* (%)	695 (5.1)	415 (4.6)	280 (6.0) *	238 (4.5)	457 (5.4) *
Diabetes, *n* (%)	423 (3.1)	201 (2.2)	222 (4.7) *	118 (2.2)	305 (3.6) *
HBP ^d^, *n* (%)	1185 (8.6)	545 (6.0)	640 (13.6) *	144 (2.7)	1041 (12.4) *
Dyslipidemia, *n* (%)	2067 (15.1)	1192 (13.2)	875 (18.7) *	341 (6.4)	1726 (20.6) *
Family obesity, *n* (%)	2310 (16.8)	1533 (17.0)	777 (16.6)	952 (17.8)	1358 (16.2) *
Family diabetes, *n* (%)	3924 (28.6)	2622 (29.0)	1302 (27.8) *	1426 (26.7)	2498 (29.8) *
Family HBP ^c^, *n* (%)	6387 (46.5)	4489 (49.7)	1898 (40.5) *	2051 (38.4)	4336 (51.7) *
Family dyslipidemia, *n* (%)	5744 (41.9)	4131 (45.7)	1613 (34.4) *	2129 (39.9)	3615 (43.1) *
Depress, *n* (%)	5256 (38.3)	3760 (41.6)	1496 (31.9) *	2538 (47.6)	2718 (32.4) *
Smoking habit, *n* (%)	2497 (18.2)	1501 (16.6)	996 (21.2) *	1040 (19.5)	1457 (17.4) *
PCS12 ^e^, mean (SD)	53.67 (6.8)	53.63 (7.0)	53.76 (6.3)	54.71 (6.4)	53.11 (7.0) *
MCS12 ^f^, mean (SD)	43.59 (10.7)	42.73 (10.9)	45.34 (10.2) *	41.05 (11.3)	44.97 (10.2) *
Rewarded Survey, *n* (%)	3942 (28.7)	2170 (24.0)	1772 (37.8) *	1938 (36.3)	2004 (23.9) *

Threshold significance was set at *p* < 0.05. * *p* < 0.05, *t*-test was used to compare the mean of continuous variables. Chi-square (χ^2^ test) was used to compare categorical variables. Work was significant by age but not by sex. ^a^ Sex: 45 people did not specify their sex. ^b^ BMI: Body Mass Index (kg/m^2^), categorized by World Health Organization (WHO) criteria. ^c^ Other includes Africans, Asians and other ethnicities. ^d^ High Blood Pressure. ^e^ Physical Component Summary. ^f^ Mental Component Summary.

**Table 2 ijerph-20-00767-t002:** The characteristics of the participants in the NUTRiMDEA study according to the lifestyle distributed by sex and age.

		Sex		Age	
	Overall	Female	Male	<40 Years	>40 Years
*n*	13,721	9031	4690	5335	8386
MDS14 ^a^, mean (SD)	7.48 (2.2)	7.61 (2.1)	7.24 (2.3) *	7.02 (2.1)	7.78 (2.1) *
Mean meals, *n* (%) >3 meals per day	6396 (46.6)	4625 (51.2)	1771 (37.8) *	2702 (50.7)	3694 (44.1) *
Snacking habit, *n* (%)	6652 (48.5)	4482 (49.7)	2170 (46.3) *	3013 (56.5)	3639 (43.4) *
Water, *n* (%) ≥7 glasses/day	4450 (32.4)	2961 (32.8)	1489 (31.8)	1956 (36.7)	2494 (29.8) *
Added salt, *n* (%) sometimes or usually	3310 (24.1)	2066 (22.9)	1244 (26.5) *	1527 (28.6)	1783 (21.3) *
Nap habit, *n* (%)	4306 (31.4)	2474 (27.4)	1832 (39.1) *	1479 (27.7)	2827 (33.7) *
Sleep weekdays, *n* (%) ≥7 h/day	8811 (64.2)	5947 (65.9)	2864 (61.1) *	3798 (71.2)	5013 (59.8) *
Sleep weekends, *n* (%) ≥7 h/day	11,692 (85.2)	7818 (86.6)	3874 (82.6) *	4656 (87.3)	7036 (83.9) *
PA ^b^ light (min/week), median [IQR]	225.0 [75.0, 270.0]	225.0 [75.0, 270.0]	180.0 [90.0, 270.0]	180.0 [75.0, 270.0]	225.0 [90.0, 270.0] *
PA moderate (min/week), median [IQR]	45.0 [0.0, 135.0]	30.0 [0.0, 90.0]	45.0 [0.0, 135.0] *	45.0 [0.0, 135.0]	30.0 [0.0, 135.0] *
PA intense (min/week), median [IQR]	90.0 [0.0, 180.0]	45.0 [0.0, 135.0]	90.0 [0.0, 225.0] *	90.0 [0.0, 180.0]	45.0 [0.0, 180.0] *
Total PA (METs-min/week), median [IQR]	1687.5 [890.3, 2691.0]	1525.5 [774.0, 2502.0]	1917.0 [954.0, 3051.0] *	1737.0 [891.0, 2722.5]	1671.0 [879.0, 2682.0] *
Time sitting, *n* (%) *					
<4 h	3472 (25.4)	2195 (24.3)	1277 (27.3)	1360 (25.5)	2112 (25.2)
5–7 h	4510 (32.9)	2938 (32.6)	1572 (33.6)	1534 (28.8)	2976 (35.5)
>8 h	5714 (41.7)	3886 (43.1)	1828 (39.1)	2430 (45.6)	3284 (39.2)
Obesogenic Score, mean (SD)	1.59 (0.91)	1.59 (0.92)	1.59 (0.90)	1.67 (0.91)	1.54 (0.91) *

Normal distributed variables expressed as mean (standard deviation) and non-normal distributed variables as median [interquartile interval]. Threshold significance was set at *p* < 0.05. * *p* < 0.05, *t*-test was used to compare the mean of continuous variables. Chi-square (χ^2^ test) was used to compare categorical variables. Differences for PA light, moderate, intense and total PA by Kruskal-Wallis test. ^a^ Mediterranean Diet Score 14-item. ^b^ Physical Activity.

**Table 3 ijerph-20-00767-t003:** Phenotypic characteristics, lifestyle, HRQoL and Obesogenic Score of the participants in the NUTRiMDEA study stratified by physical activity and diet categories analyzing for statistical interactions.

	Low PA, Low MDS14	High PA, Low MDS14	Low PA, High MDS14	High PA, High MDS14	*p* for PA	*p* for MDS14	*p* of Interaction
*n*	4867	4049	1707	2797			
Weight, mean (SE)	71.42 (0.18) ^c^	70.08 (0.19) ^b^	70.18 (0.28) ^b^	68.96 (0.23) ^a^	0.064	<0.001	0.759
Height, mean (SE)	169.9 (0.11) ^a^	169.9 (0.12) ^a^	170.1 (0.17) ^ab^	170.4 (0.14) ^b^	<0.001	<0.001	0.220
BMI, mean (SE)	24.63 (0.06) ^c^	24.16 (0.06) ^b^	24.15 (0.09) ^b^	23.65 (0.07) ^a^	<0.001	<0.001	0.815
Trousers size, mean (SE)	41.08 (0.06) ^c^	40.61 (0.07) ^b^	41.05 (0.10) ^c^	40.36 (0.08) ^a^	<0.001	<0.001	0.112
Obesity, *n* (%)	304 (6.2)	198 (4.9)	73 (4.3)	87 (3.1)	<0.001	<0.001	0.787
Diabetes, *n* (%)	154 (3.2)	143 (3.5)	46 (2.7)	54 (1.9)	0.428	0.001	0.170
HBP, *n* (%)	423 (8.7)	325 (8.0)	159 (9.3)	231 (8.3)	0.229	0.412	0.868
Dyslipidemia, *n* (%)	754 (15.5)	525 (13.0)	307 (18.0)	426 (15.2)	<0.001	0.059	0.387
MDS14, mean (SE)	6.20 (0.02) ^a^	6.49 (0.02) ^b^	9.46 (0.03) ^c^	9.62 (0.03) ^d^	<0.001	<0.001	<0.01
PA light (min/week), mean (SE)	129.0 (2.57) ^a^	287.0 (2.69) ^c^	149.0 (3.99) ^b^	301.0 (3.22) ^d^	<0.001	<0.001	0.333
PA moderate (min/week), mean (SE)	36.2 (1.41) ^a^	116.2 (1.47) ^b^	42.1 (2.18) ^a^	122.8 (1.77) ^c^	<0.001	<0.01	0.827
PA intense (min/week), mean (SE)	37.9 (1.65) ^a^	179.8 (1.73) ^b^	44.2 (2.57) ^a^	191.9 (2.08) ^c^	<0.001	<0.001	0.114
Total METs (METs-min/week), mean (SE)	874.0 (13.9) ^a^	2852.0 (14. 6) ^c^	1012.0 (21.6) ^b^	3019.0 (17.5) ^d^	<0.001	<0.001	0.316
PCS12, mean (SE)	51.67 (0.11) ^a^	53.54 (0.12) ^c^	52.33 (0.17) ^b^	54.54 (0.14) ^d^	<0.001	<0.001	0.171
MCS12, mean (SE)	41.3 (0.18) ^a^	43.6 (0.19) ^b^	43.4 (0.28) ^b^	45.4 (0.23) ^c^	<0.001	<0.001	0.547
Obesogenic Score, mean (SE)	1.90 (0.02) ^d^	1.41 (0.02) ^b^	1.63 (0.02) ^c^	1.27 (0.02) ^a^	<0.001	<0.001	<0.001

High or low physical activity as well as high or low MDS14 were determined with the median. *p*-values for two-way analysis of variance (ANOVA) with sidak post-hoc test between groups. Adjusted by type of survey, HRQoL (PCS12 and MCS12), age and sex. ^abcd^ means with different superscripts are statistically different (threshold significance was set at *p* < 0.05).

**Table 4 ijerph-20-00767-t004:** The phenotypic characteristics, HRQoL and lifestyle of participants in the NUTRiMDEA study stratified by metabolic diseases.

	Obesity		Diabetes		High Blood Pressure		Dyslipidemia	
	No	Yes	No	Yes	No	Yes	No	Yes
*n*	13,026	695	13,298	423	12,536	1185	11,654	2067
Family disease, *n* (%)	1971 (15.1)	339 (48.8) *	3654 (27.5)	270 (63.8) *	5500 (43.9)	887 (74.9) *	4297 (36.9)	1447 (70.0) *
PCS12, mean (SD)	53.9 (6.6)	49.1 (8.7) *	53.8 (6.7)	49.3 (8.4) *	54.0 (6.6)	50.2 (8.1) *	54.1 (6.5)	51.7 (7.7) *
MCS12, mean (SD)	43.7 (10.7)	42.0 (11.4) *	43.6 (10.7)	44.9 (11.3) *	43.4 (10.7)	45.4 (10.6) *	43.5 (10.7)	44.0 (10.6)
MDS14, mean (SD)	7.5 (2.1)	6.8 (2.3) *	7.5 (2.2)	6.9 (2.3) *	7.5 (2.2)	7.5 (2.2)	7.5 (2.2)	7.6 (2.2) *
Sedentarism (>8 h) *n* (%)	5424 (41.7)	290 (41.8)	5563 (41.9)	151 (35.9) *	5305 (42.4)	409 (34.6) *	4886 (42.0)	828 (40.1)
PA (METs-min/w), median [IQR]	1722.0 [891.0, 2691.0]	1251.0 [477.0, 2394.0] *	1705.5 [891.0, 2691.0]	1557.0 [688.5, 2754.0]	1708.5 [891.0, 2691.0]	1551.0 [742.5, 2573.3] *	1737.0 [891.0, 2722.5]	1525.5 [804.0, 2457.0] *
ObS, mean (SD)	1.58 (0.91)	1.74 (0.92) *	1.59 (0.91)	1.61 (0.91)	1.59 (0.91)	1.58 (0.92)	1.59 (0.91)	1.59 (0.91)
BMI (Kg/m^2^), mean (SD)	23.8 (3.4)	29.2 (3.1) *	24.0 (3.5)	25.9 (3.7) *	23.8 (3.5)	26.6 (3.5) *	23.9 (3.5)	25.2 (3.5) *

Normal distributed variables expressed as mean (standard deviation) and non-normal distributed variables as median [interquartile interval]. Threshold significance was set at *p* < 0.05. * *p* < 0.05, *t*-test was used to compare the mean of continuous variables. Chi-square (χ^2^ test) was used to compare categorical variables. Differences for total PA by Kruskal-Wallis test.

**Table 5 ijerph-20-00767-t005:** Logistic regression analysis between different diseases status (obesity, diabetes, HBP, dyslipidemia as dependent variables) and sociodemographic, family background, HRQoL, dietary and lifestyle factors.

	Obesity (*n* = 695)		Diabetes (*n* = 423)		High Blood Pressure (*n* = 1185)		Dyslipidemia (*n* = 2067)	
	Estimate	*p* Value	Estimate	*p* Value	Estimate	*p* Value	Estimate	*p* Value
Sex (female)	−0.220	0.069	−0.656	<0.001	−1.071	<0.001	−0.591	<0.001
Age (>40 years)	0.504	<0.001	0.454	<0.01	1.377	<0.001	1.242	<0.001
Survey (RS)	0.781	<0.001	0.881	<0.001	0.430	<0.001	−0.074	0.419
Family disease *	1.867	<0.001	1.215	<0.001	1.526	<0.001	1.714	<0.001
PCS12	−0.060	<0.001	−0.052	<0.001	−0.059	<0.001	−0.044	<0.001
MCS12	−0.014	<0.05	−0.006	0.409	−0.003	0.555	−0.009	<0.01
MDS14	−0.074	<0.05	−0.035	0.337	0.026	0.240	−0.014	0.419
Sedentarism (>8 h/d)	0.162	0.168	0.0241	0.872	−0.257	<0.01	−0.083	0.232
METs-min/week	0.000	0.934	0.000	0.872	0.000	0.348	0.000	0.059
Smoke (Yes)	−0.080	0.5611	0.212	0.185	0.070	0.516	0.260	<0.01

* Family disease refers to each disease of the same column. Logistic regression analysis was used to examine the association of each variable with the presence or absence of each metabolic disease (obesity, diabetes, HBP and dyslipidemia). Threshold significance was set at *p* < 0.05.

## Data Availability

Not applicable.

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
