# Peer review of "Lifestyle and Health-Related Quality of Life Relationships Concerning Metabolic Disease Phenotypes on the Nutrimdea Online Cohort"

_ijerph, 2022, doi:10.3390/ijerph20010767_

Round 1

Reviewer 1 Report

Comments

Abstract

  1. Please shuffle 2nd lane as 1st lane and 1st lane to second lane.
  2. Authors need to define HRQoL
  3. Accurate P values will be recommended in the abstract
  4. Define obesity (https://www.ncbi.nlm.nih.gov/pmc/articles/PMC8017326/) diabetes (https://www.ncbi.nlm.nih.gov/pmc/articles/PMC8017326/)  and cardiovascular disease (https://www.mdpi.com/1660-4601/18/4/1773)
  5. Write the global prevalence of NCDs and specifically in Spain also
  6. Lane 54: authors need to define metabolic syndrome (https://www.mdpi.com/1660-4601/18/4/1773)
  7. Lane-72, authors have defined WHO without expanding and in lane-81 authors have mentioned WORLD HEALTH ORGANIZATION without abbreviated form.
  8. How many days does the authors have performed physical activity and diet? Please elaborate in detail.
  9. In results section, the authors should have added the Hb1Ac levels.
  10. Discussion was seeming to be fine but authors should compare the current study results with global study results as well as with meta-analysis studies

Reviewer 2 Report

The authors conducted a cross-sectional analysis of data from the Nutrimdea Online Cohort, to characterize phenotypic and lifestyle roles on the occurrence of metabolic diseases, and to determine the potential mutual interactions and with HRQoL. Overall, the study was well executed and the results appropriately interpreted. The manuscript is well written.

I only have several comments:

Introduction

1. Line 43 “sex and gender are important determinants that impact cardiometabolic features”?

Results

2. Tables 1, 2 and 4: the SD of PA light, moderate, intense and total PA are very large. It is recommended to change them to median and interquartile intervals.

3. Figure 2: the serial numbers of several small figures are inconsistent with the description.

Discussion

4. Page 13: “the female sex, age, BMI, ... were associated with lower HRQoL scores”. The bigger or smaller the Age and BMI were related to lower HRQoL scores?

5. The prevalence of obesity, diabetes, HBP, and dyslipidemia comes from the self-report of the participants. Some people who feel physically healthy may underestimate their prevalence without going to the physical examination, thus causing greater bias. How to estimate and solve?

Reviewer 3 Report

Major comments

1. This study is based on a questionnaire, therefore, it is necessary to demonstrate how well the obtained results match the actual health survey results. Discussion is needed to compare the results of this study with data such as those from the Spanish National Health Survey. Are the prevalence of obesity, hyperlipidemia, diabetes and dyslipidemia consistent with the results of the Spanish Health Survey?

2. In a questionnaire-based survey, this study shows that more active people have more Mediterranean diets. It is necessary to compare the data that actually measure the amount of activity.

3. Are active people more conscious of health maintenance? Are people who prefer a Mediterranean diet more active? What populations are more active and less Mediterranean diets? What populations have a high Mediterranean diet but low activity? Is it the difference in intelligence or economic power?

4. Are there regional differences in the tendencies of research subjects?

5. The introduction is too long, so please make it a little more concise.

Round 2

Reviewer 3 Report

The authors sincerely responded to the reviewers' comments. This paper deserves acceptance.

Author Response

Thanks for your comments